# Characterization of Beef Coming from Different European Countries through Stable Isotope (H, C, N, and S) Ratio Analysis

**DOI:** 10.3390/molecules28062856

**Published:** 2023-03-22

**Authors:** Luana Bontempo, Matteo Perini, Silvia Pianezze, Micha Horacek, Andreas Roßmann, Simon D. Kelly, Freddy Thomas, Katharina Heinrich, Claus Schlicht, Antje Schellenberg, Jurian Hoogewerff, Gerhard Heiss, Bernhard Wimmer, Federica Camin

**Affiliations:** 1Fondazione Edmund Mach, Via E. Mach, 1, 38098 San Michele all’Adige, Italy; 2AIT Austrian Institute of Technology GmbH, 2444 Seibersdorf, Austria; 3Isolab GmbH, Woelkestr. 9/1, 85301 Schweitenkirchen, Germany; 4Food Safety & Control Laboratory, Joint FAO/IAEA Centre of Nuclear Techniques in Food and Agriculture, International Atomic Energy Agency, Vienna International Centre, Wagramer Strasse 5, P.O. Box 100, 1400 Vienna, Austria; 5Eurofins Analytics France, Authenticity Competence Centre, Rue P.A. Bobierre, 44323 Nantes, France; 6Fera Science Ltd., Sand Hutton, York Y041 1LZ, UK; 7LGL Bayerisches Landesamt für Gesundheit und Lebensmittelsicherheit, Veterinärstraße 2, 85764 Oberschleißheim, Germany; 8National Centre for Forensic Studies, Faculty of Science and Technology, University of Canberra, Canberra 2617, Australia; 9AIT Austrian Institute of Technology GmbH, Giefinggasse 4, 1210 Vienna, Austria

**Keywords:** geographical origin, authenticity, food fraud, beef, meat, stable isotope ratio analysis

## Abstract

The need to guarantee the geographical origin of food samples has become imperative in recent years due to the increasing amount of food fraud. Stable isotope ratio analysis permits the characterization and origin control of foodstuffs, thanks to its capability to discriminate between products having different geographical origins and derived from different production systems. The Framework 6 EU-project “TRACE” generated hydrogen (^2^H/^1^H), carbon (^13^C/^12^C), nitrogen (^15^N/^14^N), and sulphur (^34^S/^32^S) isotope ratio data from 227 authentic beef samples. These samples were collected from a total of 13 sites in eight countries. The stable isotope analysis was completed by combining IRMS with a thermal conversion elemental analyzer (TC/EA) for the analysis of *δ*(^2^H) and an elemental analyzer (EA) for the determination of *δ*(^13^C), *δ*(^15^N), and *δ*(^34^S). The results show the potential of this technique to detect clustering of samples due to specific environmental conditions in the areas where the beef cattle were reared. Stable isotope measurements highlighted statistical differences between coastal and inland regions, production sites at different latitudes, regions with different geology, and different farming systems related to the diet the animals were consuming (primarily C3- or C4-based or a mixed one).

## 1. Introduction

Frequent problems with food safety and quality can result in a loss of consumer confidence and multi-million euro losses for sectors such as the meat industry [1]. Governments have highlighted the need for stricter controls on food safety, authenticity, and geographical origin. The use of traceability systems to go back through a supply chain from ‘fork to farm’ represents a significant challenge to allow the identification of sources of contamination, breaches in quality, and fraudulent or accidental misdescription [1].

Documents and labeling may become unreliable when the origin is intentionally mislabeled. In this case, an investigation of the product itself is required to identify and control the real production origin. Stable isotope ratio analysis (SIRA) is arguably one of the most powerful tools for this purpose. This technique is already routinely applied for the control of authenticity of wine produced in EU countries [2,3] and has been proven to be successful for many other kinds of foodstuff [4,5]. Several studies on meat [6], such as lamb [7,8], pork [9,10], and chicken [11] have already been reported.

It has already been demonstrated that beef cattle reared in Europe can be distinguished from beef produced in North and South America [12,13]. Similarly, Korean and Japanese beef can be differentiated from American and Australian beef [14,15,16]. The isotope ratios δ(^13^C), δ(^15^N), δ(^34^S), δ(^2^H), and δ(^18^O) analyzed by Zhang et al. in samples from Argentina, Brazil, Canada, China, New Zealand, and Uruguay enabled satisfactory classification using discriminatory analysis with an original classification accuracy of 96.6% and a cross-validation accuracy of 95.9% [17]. However, to the best of our knowledge, only a few studies have considered European samples coming from neighboring countries [18,19], leaving a gap concerning the applicability of this method to verify the origin of meat from beef cattle reared within the European region. 

The carbon (*δ*(^13^C)), nitrogen (*δ*(^15^N)), and sulphur (*δ*(^34^S)) isotope ratios of the meat are mainly related to the feed the animals have been consuming [6]. The *δ*(^13^C) depends on the photosynthetic cycle of the plants included in the animal diet [20]. Whilst the *δ*(^15^N) and *δ*(^34^S) are influenced by factors like the agricultural practices that have been applied to the soil on which the animals have been grazing or where the plants that are incorporated into composite feed or fodder grew (e.g., the use of fertilizers). Furthermore, the inclusion of particular fodder ingredients can influence the isotopic ratios of the feed (e.g., *δ*(^15^N) effected by leguminous plants, seaweed, fishmeal, and amino acids) [6]. The *δ*(^34^S) values are also related to the geology of the soil and are therefore useful parameters to be considered in the geographical characterization of animal products [21]. The hydrogen (*δ*(^2^H)) isotope ratio of the animal tissues reflects the isotopic composition of the water that an animal consumed, whether directly or through moisture in the feed. As the isotopic composition of water varies regionally due to fractionation in the global hydrological cycle, these variations are also incorporated into the meat [12,22]. 

This investigation has been undertaken within the European project Tracing Food Commodities in Europe (TRACE) (FP6–2003–FOOD–2–A), which has already demonstrated how SIRA of light elements can be applied to control different products: lamb meat [7], mineral water [23], olive oil [24], honey [25], and wheat [26] (the latter also including ^87^Sr/^86^Sr data). In this paper, we present the isotope values (*δ*(^13^C), *δ*(^15^N), *δ*(^34^S), and *δ*(^2^H)) of beef coming from cattle reared at 13 sites spread throughout 8 European countries, considering the useful outcomes, as well as the limitations, of stable isotope analysis to determine and control the declared geographical origin. The measurement of the strontium isotope ratios (^87^Sr/^86^Sr) of most of the beef samples in the present work has been reported previously by Rummel et al. [27].

## 2. Results and Discussion

The mean isotopic ratios and standard deviation of *δ*(^13^C), *δ*(^15^N), *δ*(^34^S), and *δ*(^2^H) for the 8 countries involved in the study and for each of the 13 sampling points are reported in Table 1.

### 2.1. Carbon Stable Isotopic Ratio

The carbon isotope composition of beef is mainly influenced by the cattle’s feed, which is composed of plants and, in some cases, concentrates and supplements. The *δ*(^13^C) of the beef muscle tissue, which has come into equilibrium with the diet, is therefore ultimately influenced by the photosynthetic pathway that the cattle-diet plants utilize for CO_2_ fixation [20]. On this basis, plants can be categorized into three groups: C3, C4, and CAM plants [28]. The C3 plants, representing about 85% of the plant species on the planet, fix carbon via the Calvin cycle through the Rubisco enzyme (ribulose-1,5-biphosphate carboxylase oxygenase) [29]. These plants have values ranging between −33 and −23‰ [28]. On the other hand, both C4 and CAM plants follow alternative photosynthetic pathways to fix CO_2_, adapted to improve the efficiency of photosynthesis in hot and arid environments and reduce water loss through stomatal evapotranspiration. The C4 plants fix the CO_2_ through a non-Rubisco enzyme, producing a 4-carbon organic acid (oxaloacetic acid), from which their name derives [29]. Their *δ*(^13^C) values range between −14 and −12‰ [28]. Finally, CAM plants, which possess *δ*(^13^C) values ranging between C3 and C4 values, use a different water-saving pathway called crassulacean acid metabolism (CAM), based on the temporal separation of CO_2_ fixation and the subsequent sugar synthesis [29].

Forages and feed concentrates included in the animals’ diets may be derived from both C3 and C4 plants. However, in Europe, C4-derived values in bovine muscle tissue *δ*(^13^C) are synonymous with the feeding of maize (Zea mays or corn), as this is the only C4 fodder crop of commercial importance in the countries where the test samples originated. 

In this study, the *δ*(^13^C) values of the bovine muscle protein (defatted dry mass) ranged from −15.5‰ to −27.6‰. As a result of the ANOVA, the three groups considered in Table 1 were statistically different (*p* < 0.05). The C_LL_C4 group (*δ*(^13^C) = −19.3 ± 1.5‰) includes Mediterranean countries (Greece, Italy, and Spain), in which the substantial use of C4 plant-based concentrated feed in the rearing practices is widespread. However, the mean value, together with the standard deviation, evidence different feeding practices, which at least partially might be related to intensive (in a stable, use of feed with C4-plant (maize) dominance) or extensive (rearing on meadows, with only minor maize feed additions to grazing, if at all) rearing of cattle. Our results are consistent with the *δ*(^13^C) values obtained by Osorio et al. for Italian and Spanish samples [18,19]. 

On the other hand, northern coastal regions seem to have predominantly grazing systems based on C3 pasture plants, with the C_ML_C3 group having a mean *δ*(^13^C) of −26.5 ± 0.8‰. These findings are consistent with published *δ*(^13^C) data from sampling points located in the United Kingdom and Ireland [18,19].

For the remaining group I_LL_MIX, including alpine and mountainous areas, there is a wide variety of grazing systems with intermediate *δ*(^13^C) results and a relatively higher standard deviation (*δ*(^13^C) = −22.7 ± 2.9‰). The results are consistent with other studies carried out on beef samples coming from France, Austria, and Germany [18,19]. Nevertheless, in this group, areas that seem to be based mainly on a C3 diet, such as Mühlviertel (A) and the Allgäu (DE), can be readily identified. However, the high standard deviation in *δ*(^13^C) for almost all the regions in this group demonstrates the use of highly variable rearing practices. 

#### Carbon Isotopic Ratio of the Beef Fat

The *δ*(^13^C) of the fat and the protein fraction of meat are known to be correlated, as reported in the literature [30,31,32]. Nevertheless, to exclude the lack of additional information derived from the analysis of the *δ*(^13^C) of the fat fraction, a subset of beef muscle samples was measured to assess this parameter.

The *δ*(^13^C)_FAT_ of all Italian samples (Sicily, Trentino, and Tuscany) was isolated and compared to the *δ*(^13^C)_PROTEIN_. A mean depletion of 4.1 ± 1.1‰ in fat compared to the protein samples was calculated, in line with the findings of other authors in other animal species [33,34,35]. As expected, *δ*(^13^C)_FAT_ and *δ*(^13^C)_PROTEIN_ are correlated with a linear fit (*δ*(^13^C)_FAT_ = −0.14 + 1.19 *δ*(^13^C)_PROTEIN_, r^2^ = 0.88, *p* < 0.001). For the purpose of this study, as the analysis of the fat provides no additional information regarding its origin, it will be excluded from further discussion.

### 2.2. Nitrogen Stable Isotopic Ratio 

The *δ*(^15^N) of beef is mainly affected by the species of forage plants eaten by the cattle and the nitrogen pool in the soil where the forage plants or pasture grow. Leguminous plants (e.g., pea, clover, soybean) can incorporate nitrogen directly from their rhizo-symbionts that metabolize the atmospheric nitrogen, resulting in relatively low isotopic ratios (around 0‰) [8]. For non-nitrogen-fixing plants, their *δ*(^15^N) is mainly influenced by the nitrogen isotope composition of the water-soluble soil nitrate and fertilizers [36]. In particular, synthetic fertilizers, produced from atmospheric nitrogen via the Haber process, have *δ*^15^N values between −4‰ and +4‰, while organic fertilizers are characterized by values between +0.6‰ and +36.7‰ [37,38]. Moreover, highly elevated *δ*(^15^N) values may be indicative of the inclusion of marine plants and fishmeal, enriched in ^15^N, in the feed or as fertilizer ingredients [39].

In this study, *δ*(^15^N) of the beef protein fraction ranged from 3.0‰ up to 9.6‰. The results we obtained are consistent with *δ*(^15^N) obtained for beef samples coming from various European countries [18,19]. Coastal areas such as Barcelona (ES) (*δ*(^15^N) = 6.8 ± 0.5‰), Orkney (UK) (*δ*(^15^N) = 7.7 ± 1.0‰), and Cornwall (UK) (*δ*(^15^N) = 6.0 ± 1.2‰) showed relatively higher and more consistent *δ*(^15^N) than the rest of the dataset, which possessed a low standard deviation (*δ*(^15^N) = 5.4 ± 1.1‰). As previously mentioned, this may be due to the use of organic fertilizers or products of the marine ecosystem in the agricultural practices of the area. For Orkney, Ireland, and Cornwall, a further explanation might be the rearing practice: extensive rearing of the cattle on meadows for (almost) the entire year might have led to an enrichment in the *δ*^15^N of the soil and thus the meadows the cattle foraged on due to the cattle manure, in contrast to cattle reared in stables and fed on corn and cereals fertilized with synthetic fertilizers. This also might explain the *δ*^15^N values of the German and Austrian beef samples because, even if the cattle there graze on meadows in the summer, they are kept in a stable and fed on dry feed (that at least partially might have been produced with the application of synthetic fertilizers) during the rest of the year. Particularly low results such as those of Chalkidiki (GR) (*δ*(^15^N) = 4.3 ± 1.6‰) and Sicily (IT) (*δ*(^15^N) = 4.3 ± 1.6‰) may be due to the dominant use of feed produced with synthetic fertilizers and/or to an increased amount of Leguminosae among concentrated feed or foraged plants.

### 2.3. Sulphur Stable Isotopic Ratio

The *δ*(^34^S) of beef muscle protein tends to reflect the values of the plants the cattle were fed on. In turn, the *δ*(^34^S) values of the plants are mainly influenced by the geology of the soil the plant grew on (for instance, the presence of sulfates or sulfides in the soil and the type of underlying local bedrock). Other factors of influence, such as fertilization practices that include sulphur amendment [21,40,41] and proximity to the sea (the sea-spray effect) [21,40], as present-day seawater has a *δ*(^34^S) value of approximately 21–22‰ [42], also have to be considered.

In the present study, *δ*(^34^S) ranged from −1.7 up to 14.5‰. In general, coastal regions show relatively elevated *δ*(^34^S) values compared to the rest of the dataset. This may be the result, as previously mentioned, of the sea-spray effect on pastures where cattle are grazing, or in the fertilization practices of products coming from the marine ecosystem, or of the use of fertilizers rich in sulphur and having high *δ*(^34^S) values, which may be produced from salt deposits of earlier geological times (evaporites) [43]. The influence of sea-spray may be very variable between different sites; e.g., Mizota and Sasaki [42] reported notably lower *δ*(^34^S) values only 16km inland from the sea in Japan. Thus, we assume a dominant sea-spray effect for the samples from Orkney, where all pasture areas are within 16km of the coast. For the Irish samples, a minor influence of sea-spray can be assumed, as the lowest *δ*(^34^S) values measured in Irish sheep wool [44] (reporting sheep *δ*(^34^S) values from various localities on the island) are around 6–7‰ and maximum values of ca. 17‰ at the western and below 11‰ at the eastern coast. Bearing in mind these maximum values, the *δ*(^34^S) data of Orkney beef samples are surprisingly low, even more so as the average elevation of Orkney is around 100 m (or slightly less). This suggests that the bedrock geology might have a notable influence.

Nevertheless, Florence and Sicily represent an exception, having *δ*(^34^S) values (4.4 ± 0.9‰ and 3.3 ± 1.6‰, respectively) lower than the rest of the previously mentioned coastal regions. This may be due to the volcanic nature of the soil, which characterizes the Italian regions of Tuscany and Sicily, resulting in relatively low *δ*(^34^S) [45]. However, volcanism as an explanation for low or high *δ*(^34^S) values [46] has been challenged, and the geological map of Tuscany shows just very small volcanic areas, so the oxidation of sedimentary sulfides is a more plausible explanation for the observed low *δ*(^34^S) values [42]. Particularly low values are also reported by Osorio et al. [47] for beef samples coming from Italy (*δ*(^34^S) = 1.5 ± 2.3‰) and by Camin et al. [7] for lamb samples coming from Sicily (*δ*(^34^S) = 2.5 ± 1.7‰) and Tuscany (*δ*(^34^S) = 3.8 ± 0.6‰ for lambs given a mixed diet; *δ*(^34^S) = 1.9 ± 0.7‰ for milk-fed lambs).

As for the I_LL_MIX group, relatively low values (*δ*(^34^S) = 5.0 ± 1.5%) can be attributed to the *δ*(^34^S) of the bedrock geology, confirming that the geology is the most relevant factor in determining the *δ*(^34^S) of most terrestrial animal food products. The sites of Frankonia, Allgäu (DE), and Mühlviertel (A) agree with this interpretation, having relatively low values (*δ*(^34^S) = 4.1 ± 1.7‰, *δ*(^34^S) = 3.3 ± 1.4‰, and *δ*(^34^S) = 3.9 ± 0.1‰, respectively). The just-mentioned results for the German sites are consistent with previous values reported by Auerswald et al. for the hair of cattle reared in Grünschwaige (DE) [48]. Particularly low values for samples coming from different parts of Austria are reported in the literature [47].

### 2.4. Hydrogen Stable Isotopic Ratio

The isotopic composition of animal meat is related to the feed and water that the animal ingests [7,18]. The feed has previously been shown to be the main source of animal muscle protein hydrogen [49,50]. Nevertheless, Perini et al. proved that even though only around 30% of hydrogen body protein derives from drinking water [49,50], the H isotopic composition of defatted dry mass records the deuterium signature of meteoric water [8].

The isotopic ratio of the water (H− and O−isotopes) depends on the water source the cattle are drinking, whether groundwater (tap water), surface water, or water in the feed (e.g., water in pasture grass). The precipitation hydrogen and oxygen *δ*(^18^O) isotopic ratios are highly variable because of two factors: the temperature effect (the cooler the air temperatures, the lower the isotopic composition of precipitation) and the continental effect (a greater distance from the sea causes a depletion in the isotopic composition of the precipitation) [51,52] through the “rain-out” of the heavier isotopologues of water as clouds move inland. In this way, lowlands close to the sea and warm regions are characterized by water enriched in the heavy isotopes of hydrogen and oxygen (^2^H and ^18^O). On the other hand, regions far from the sea, at high altitudes, and with low temperatures have water with relatively depleted *δ*(^2^H) isotope values. Moreover, due to plant evapotranspiration, the water in the fresh cattle feed is significantly enriched in its isotopic composition with respect to the water the plant takes up from the soil [53]. 

In this study, the *δ*(^2^H) values of the bovine protein ranged from −78.2‰ to −126.0‰. According to the ANOVA, the *δ*(^2^H) of the three groups considered in this study were statistically different (*p* < 0.05). The C_LL_C4 group, including samples coming from coastal regions sited at low latitudes, has the highest values in the datasets (*δ*(^2^H) = −91.7 ± 5.6‰). In particular, samples coming from Sicily (IT) (*δ*(^2^H) = −87.0 ± 3.5‰) possessed the highest *δ*(^2^H) values in the dataset (Table 1). Despite being coastal, as the previously mentioned group, C_ML_C3 samples are characterized by higher latitudes than C_LL_C4, resulting in relatively lower values (*δ*(^2^H) = −95.1 ± 5.0‰). Finally, the lowest values have been found in the I_LL_MIX group, including samples coming from inland sites, mostly alpine or hilly/mountainous locations. This can be explained by the long distance of the cattle-rearing sites from the sea. In this group, samples from Allgäu (DE) were the lowest of all the datasets (*δ*(^2^H) = −106.5 ± 7.5‰).

As previously stated, the factor that mainly influences the *δ*(^2^H) isotopic ratio of the plants, and thus of the animal proteins, is water, whose primary source is the rainfall. Therefore, the correlation between the average *δ*(^2^H) values of beef protein and the annual average isotopic signature of rainfall has been evaluated at the 13 different sampling locations and is represented in Figure 1. In the absence of direct measurement of the *δ*(^2^H) of the rainwater, water isotope data from the WaterIsotope database administered by Gabriel Bowen have been used. The data available in the http://wateriso.utah.edu (accessed on 1 March 2023) database are monthly weighted average precipitation values for sites all over the world. When possible, the GPS coordinates of the beef cattle rearing locations have been used to calculate the predicted *δ*(^2^H) value of the water at that sampling site. The complete dataset used to perform the correlation represented in Figure 1 is reported in Appendix A. As expected, *δ*(^2^H) of the beef protein and precipitation are positively correlated (linear correlation, r^2^ = 0.77, *p* < 0.001).

### 2.5. Principal Component Analysis

The results of the PCA carried out on the variables are summarized in Figure 2, which displays the objects and the variables simultaneously projected in the space of the first two PCs. Even though no information on the sample provenance or the animal diets is incorporated into the PCA, the biplot reveals a clustering of the three groups: C_LL_C4, C_ML_MIX, and I_LL_C3.

The *δ*(^15^N) and *δ*(^34^S) values of the beef protein are almost parallel to the direction along which the C_ML_C3 group is separated from the rest of the dataset and contribute to the identification of this group with respect to the others. This observation agrees with the previously discussed interpretation of the stable isotope results. Indeed, higher *δ*(^15^N) and *δ*(^34^S) values have been found for the C_ML_C3 group with respect to the others. This is likely due to the closeness to the sea of the sampling points included in the group (Bohernagore (IE), Orkney, and Cornwall (GB)) and the respective bedrock geology, assuming a dominant influence for the latter. On the other hand, despite being coastal, samples of group C_LL_C4 are characterized by relatively low *δ*(^34^S) due to the volcanic origin of two of the sites belonging to this group (Sicily, IT) and a peculiar geology (Tuscany, IT), while samples of group I_LL_MIX resulted in relatively low both *δ*(^15^N) and *δ*(^34^S), due to their geology and, with respect to *δ*(^15^N) values, probably the feeding practice.

The separation between the C_ML_C3 group and the rest of the dataset occurs almost parallel to PC1, and *δ*(^2^H) seems to be the parameter that most contributes to this discrimination. Indeed, while group C_ML_C3 includes middle latitude samples, the other groups include low latitude samples, and *δ*(^2^H) is known to depend on this parameter. 

Finally, as for the discrimination based on the diet that the animals have been given, the separation among the three groups seems to occur along the bisector of PC1 and PC2 axes, and all the isotopes measured appear to make a contribution. Indeed, as previously mentioned, all isotopes under study are influenced by the diet that the animals have been exposed to, whether through feeding or through drinking water, reinforcing the maxim that “you are what you eat (isotopically)” [54].

## 3. Materials and Methods

### 3.1. Samples

Beef samples have been taken directly at the cattle farms or in slaughterhouses. A total of 227 samples, coming from 13 sites spread across 8 different European countries, have been considered: Bohernagore (Republic of Ireland, IE), Orkney and Cornwall (United Kingdom), Limousin (France, FR), Barcelona (Spain, ES), Frankonia, Allgäu, and Gäuboden (Germany, DE), Mühlviertel (Austria, A), Trentino, Tuscany, and Sicily (Italy, IT), and Chalkidiki (Greece, GR). Whenever possible, the GPS coordinates and elevation above sea level of each sampling point were recorded.

In discussing the results, the samples have been grouped according to the geographical origin and the diet of the animals. In particular, the first group is named I_LL_MIX, as it includes samples coming from inland areas (I) sited at relatively low latitudes (LL), where the animals have been given a mixed diet (Limousin, Frankonia, Allgäu, Gäuboden, Mühlviertel, and Trentino); similarly, group C_LL_C4 includes the samples coming from coastal areas (C) sited at relatively low latitudes (LL), where the animals have been given a C4-based diet (Barcelona, Chalkidiki, Sicily, and Tuscany); finally, group C_ML_C3 includes the samples coming from coastal areas (C) sited at higher latitudes (ML) compared to the other groups, where the animals have been given a C3-based diet (Cornwall, Orkney and Bohernagore).

### 3.2. Preparative and Analysis Procedures

The samples were frozen or cooled during transport to the respective isotope laboratories. The meat was cut, minced, and lyophilized. Afterwards, the samples were freeze-dried and defatted with a soxhlet apparatus using petroleum ether [7,55]. The fat-free residue (or dry mass), considered below as predominantly the protein fraction, was collected and saved for the C, N, S, and H analyses, while the fat fraction of a sub-set of samples was considered in paragraph 3.1.1 to make a comparison between *δ*(^13^C)_FAT_ and *δ*(^13^C)_PROTEIN_.

### 3.3. Stable Isotope Ratio Analysis

All samples were weighed into silver and tin capsules for H− and CNS−isotope measurements, respectively. The capsules were introduced into a thermal conversion elemental analyzer (TC/EA) for the analysis of *δ*(^2^H) and an elemental analyzer (EA) for the determination of *δ*(^13^C), *δ*(^15^N), and *δ*(^34^S). As the samples have been processed and measured in several laboratories, different isotope ratio mass spectrometers (IRMS) and peripheral devices have been used (see [7,25,55]).

In agreement with the IUPAC protocol, the isotopic values are expressed in ‘delta notation’ in relation to the international standards V–PDB (Vienna–Pee Dee Belemnite) for *δ*(^13^C), V–SMOW (Vienna–Standard Mean Ocean Water) for *δ*(^2^H), V–CDT (Vienna–Canyon Diablo Troilite) for *δ*(^34^S), and Air (atmospheric N_2_) for *δ*(^15^N), following equation (1):(1)$$\delta_{ref}{(}^{i}E{/}^{j}E,sample)=\left[\frac{R{(}^{i}E{/}^{j}E,\;sample)}{R{(}^{i}E{/}^{j}E,\;ref)}\right]-1$$
where *ref* is the international measurement standard, *sample* is the analyzed sample, and *^i^E*/*^j^E* is the isotope ratio between heavier and lighter isotopes [20]. The delta values are multiplied by 1000 and expressed commonly in units “per mil” (‰) or, according to the International System of Units (SI), in units milliurey (mUr) [56].

The isotopic values were calculated against two standards through the creation of a linear equation. The standards that have been used in the isotopic analyses were international reference materials or in-house working standards that have been calibrated against them. In particular, the international standards that have been used are: for ^13^C/^12^C, fuel oil NBS–22 (*δ*(^13^C) = −30.03 ± 0.05‰), sucrose IAEA–CH–6 (*δ*(^13^C) = −10.45 ± 0.04‰) (IAEA–International Atomic Energy Agency, Vienna, Austria), and L–glutamic acid USGS 40 (*δ*(^13^C) = −26.39 ± 0.04‰) (U.S. Geological Survey, Reston, VA, USA); for ^15^N/^14^N, L–glutamic acid USGS 40 (*δ*(^15^N) = −4.52 ± 0.06‰) (U.S. Geological Survey, Reston, VA, USA), ammonium sulfate salts IAEA–N–1 (*δ*(^15^N) = +0.43 ± 0.07‰) and IAEA–N–2 (*δ*(^15^N) = +20.41 ± 0.12‰) and potassium nitrate IAEA–NO3 (*δ*(^15^N) = +4.7 ± 0.2‰); for ^34^S/^32^S, USGS 42 (*δ*(^34^S) = +7.84 ± 0.25‰), USGS 43 (*δ*(^34^S) = +10.46 ± 0.22‰), Barium sulphate IAEA–SO–5 (*δ*(^34^S) = +0.5 ± 0.2‰) and NBS 127 (*δ*(^34^S) = +20.3 ± 0.4‰); for ^2^H/^1^H fuel oil NBS–22 *δ*(^2^H) = −119.6 ± 0.6‰) and Keratins CBS (Caribou Hoof Standard *δ*(^2^H) = −157 ± 2‰) and KHS (Kudu Horn Standard *δ*(^2^H) = −35 ± 1‰) from U.S. Geological Survey.

Each reference material was measured in duplicate at the start and end of each daily group of analyses of samples (each sample was also analyzed in duplicate). A control sample was also included in the analyses of each group of samples to check the efficiency of the measure. The maximum standard deviations of repeatability accepted were 0.3‰ for *δ*(^13^C) and *δ*(^15^N), of 0.4‰ for *δ*(^34^S), and of 3‰ for *δ*(^2^H).

Additional interlaboratory comparisons on meat samples have been done at the beginning of the project and throughout its duration to evaluate its accuracy and protect against bias among the seven laboratories involved in the project.

The laboratories involved and the standard deviations between labs calculated for *δ*(^13^C), *δ*(^15^N), *δ*(^34^S), and *δ*(^2^H) are reported in Appendix A.

### 3.4. Statistical Analysis

Statistical analysis was carried out using Statistica 14.0.1.25. A one–way ANOVA was used to test the effect of the geographical origin and the animal diet on the isotopic ratios, applying Tukey’s test for post-hoc analysis. The p-value cutoff was set at <0.05, indicating significant statistical differences. The linear correlations reported in Section 2.1 and Section 2.4 were performed by considering the Pearson’s coefficient and setting a p-value cutoff of <0.001.

To first explore the dataset and graphically represent all the data, a PCA (principal component analysis) was performed. The objective of data reduction methods like principal component analysis is the reduction of the number of variables and the detection of structure in the relationships between variables [57]. Patterns in a data matrix can be emphasized by projecting objects and variables into the space of a few significant PCs with minimal loss of information. The dataset of five isotope parameters has been reduced to four factors, and the first two, PC1 and PC2, explain a total of 77.5% of the data variation.

## 4. Conclusions

In this study, a total of 227 defatted muscle tissue samples derived from beef cattle reared at 13 geographical locations spread across the European continent have been collected and isotopically analyzed. The dataset has been divided into three main groups (C_LL_C4, C_ML_C3, and I_LL_MIX) having similar characteristics (based on geographical features and breeding practices) to easily characterize similar groups of samples. The SIRA of *δ*(^13^C), *δ*(^15^N), *δ*(^34^S), and *δ*(^2^H) has been applied to detect statistical differences in the dataset.

Both *δ*(^13^C) and *δ*(^2^H) gave statistically significant differences between the three groups, whether due to the different feeding regimes of the animals or to the different latitudes and climatic conditions of the sampling points, respectively. Particularly high *δ*(^15^N) values have been found in some coastal sampling points, while the rest of the dataset showed homogeneous values. Regarding *δ*(^34^S), higher values were found for coastal areas with respect to inland regions. The Italian regions of Sicily and Tuscany were an exception, probably due to the volcanic nature of their soil. To better highlight clusters in the dataset, a PCA was performed. The biplot displaying the first two PCs, explaining in total 77.5% of the data variation, reveals a clustering of the three groups C_LL_C4, C_ML_MIX, and I_LL_C3, confirming the underlying biogeoclimatic hypotheses.

In this work, stable isotope analysis was confirmed to be a powerful tool to discriminate among beef samples coming from neighboring European countries, to distinguish coastal and inland regions, and areas having different latitudes and breeding systems. In the future, it would be worth producing other chemometric models based on supervised pattern recognition, such as linear discriminant analysis (LDA), aiming to assess the origin of unknown beef samples. The present results might also help in the future construction of an isotopic databank built for each country and aiming to assess the authenticity of beef samples and potentially protect and promote PDO and PGI products, e.g., Orkney beef.

## Figures and Tables

**Figure 1 molecules-28-02856-f001:**
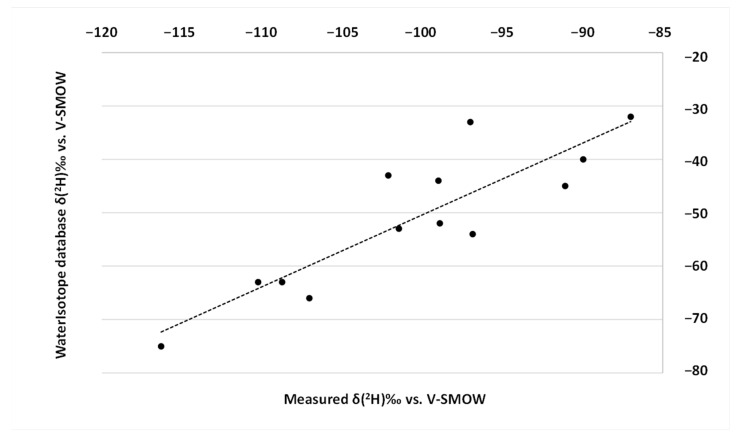
Correlation between annual δ(^2^H) of water at the beef sampling location estimated by the Water-Isotope database (Oxygen Isotopes in Precipitation Calculator) and the average δ(^2^H) values measured in the beef protein.

**Figure 2 molecules-28-02856-f002:**
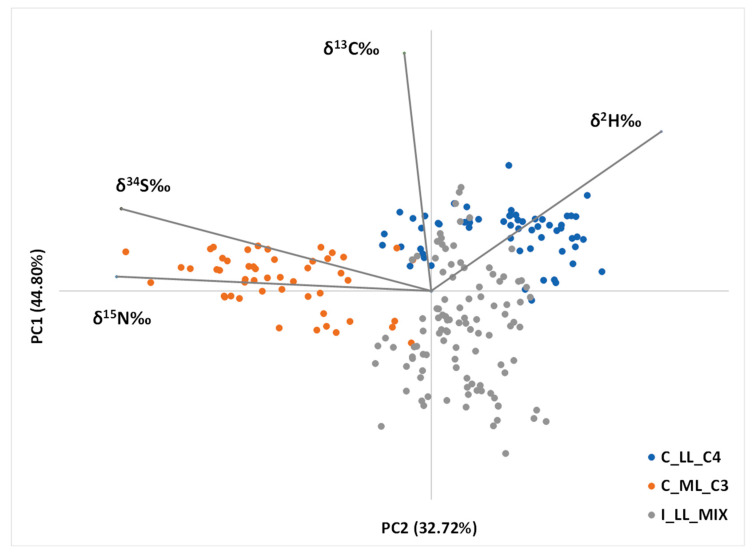
Plot showing the first two principal components (PC1 and PC2) determined from the analysis of beef protein *δ*(^2^H), *δ*(^13^C), *δ*(^15^N), and *δ*(^34^S) values. The three groups considered are identified by different colors: C_LL_C4 (animals from coastal areas, sited at relatively low latitudes, given a C4-based diet (Barcelona, Chalkidiki, Sicily, and Tuscany)) in blue; C_ML_C3 (animals from coastal areas sited at higher latitudes compared to the other groups, given a C3-based diet (Cornwall, Orkney and Bohernagore)) in orange; I_LL_MIX (animals coming from inland areas sited at relatively low latitudes, given a mixed diet (Limousin, Frankonia, Allgäu, Gäuboden, Mühlviertel, and Trentino)) in grey.

**Table 1 molecules-28-02856-t001:** Mean *δ*(^13^C), *δ*(^15^N), *δ*(^34^S), and *δ*(^2^H) of bovine muscle protein and standard deviations for the 13 European sites considered. The dataset has been divided in three groups: C_LL_C4 (animals from coastal areas, sited at relatively low latitudes, given a C4-based diet); I_LL_MIX (animals coming from inland areas sited at relatively low latitudes, given a mixed diet); C_ML_C3 (animals from coastal areas sited at higher latitudes compared to the other groups, given a C3-based diet).

Group	Country	Site (N. Samples)	*δ*(^2^H) (‰, Vs. V–SMOW)	*δ*(^13^C) (‰, Vs. V–PDB)	*δ*(^15^N) (‰, Vs. Air)	*δ*(^34^S) (‰, Vs. V–CDT)
C_LL_C4	Greece	Chalkidiki (4)	−99.0 ± 8.0	−20.3 ± 2.1	4.3 ± 1.6	6.1 ± 0.6
	Spain	Barcelona (15)	−97.0 ± 3.6	−18.4 ± 1.1	6.8 ± 0.5	7.1 ± 0.4
	Italy	Florence (20)	−91.1 ± 2.7	−18.9 ± 1.4	5.2 ± 0.6	4.4 ± 0.9
		Sicily (20)	−87.0 ± 3.5	−20.2 ± 1.3	4.4 ± 0.6	3.3 ± 1.6
I_LL_MIX		Trento (33)	−101.5 ± 4.3	−22.0 ± 1.8	4.8 ± 0.8	5.6 ± 0.5
	Austria	Mühlviertel (4)	−110.3 ± 5.0	−24.1 ± 2.6	4.7 ± 0.4	3.9 ± 0.1
	France	Limousin (20)	−98.9 ± 6.0	−21.3 ± 3.2	5.6 ± 0.6	6.6 ± 0.7
	Germany	Frankonia (20)	−108.8 ± 4.7	−22.9 ± 2.8	6.0 ± 1.0	4.1 ± 1.7
		Allgäu (20)	−116.3 ± 4.1	−25.0 ± 1.8	6.0 ± 1.2	3.3 ± 1.4
		Gäuboden (20)	−107.1 ± 3.7	−22.3 ± 3.6	6.2 ± 0.9	5.3 ± 0.5
C_ML_C3	UK	Cornwall (20)	−90.0 ± 1.9	−26.0 ± 1.1	7.7 ± 1.0	8.1 ± 1.6
		Orkney (23)	−96.9 ± 2.5	−26.8 ± 0.4	7.9 ± 0.6	10.0 ± 3.1
	Ireland	Bohernagore (8)	−102.1 ± 2.6	−26.7 ± 0.3	6.4 ± 1.0	8.8 ± 1.6

## Data Availability

The raw data presented in this study are available on request from the corresponding author.

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
