# Peer review of "Characterization of Beef Coming from Different European Countries through Stable Isotope (H, C, N, and S) Ratio Analysis"

_molecules, 2023, doi:10.3390/molecules28062856_

Round 1
Reviewer 1 Report
In many areas cattle will be fed at least part of the time with feed that may not have grown in their own area. Have the authors considered how this may affect their results and if yes, what steps were taken to ameliorate this?
Author Response
Reviewer 1:
In many areas cattle will be fed at least part of the time with feed that may not have grown in their own area. Have the authors considered how this may affect their results and if yes, what steps were taken to ameliorate this?
We thank the referee for this comment, which gives us the chance to better explain the purpose of this study. The Trace Project, to which these samples belong, aimed to map the ranges of isotopic variability of real samples of different matrices (from oil to honey and, in this study, beef). Therefore, animals reared under controlled conditions on pre-established diets were not considered, whereas commercially available beef, reared according to the practices followed in the country/locality of sampling, were the samples the study aimed to collect. The purpose of the study was therefore to photograph the real situation of European beef market, considering that certainly some of the ingredients of animal diets may not be local. The isotopic ratios of carbon, nitrogen and sulphur confirmed their close correlation with the animal C3-C4 diet regardless of the origin of the ingredients, while deuterium confirmed its correlation with the drinking water (which must be local), resulting as the most effective geographical tracer.
Reviewer 2 Report
This manuscript addresses authentication of European beef by using stable isotope analysis. The authors analyze isotope ratio of hydrogen, carbon, nitrogen and sulfur in beef of several regions. The values have been used for earth science such as resource mapping and recently apply for food science. Then, these provide chemical finger print which derives from circumstance and scientific evidence. There are various reports about authentication of beef, especially famous and expensive brands. I recommend that this paper be accepted after minor revision.
The isotope ratio of carbon, nitrogen, and hydrogen/oxygen provides difference of feed, origin of protein in feed and isotope ratio of water in growth region (for example, Rapid Commun Mass Spectrom. 2020;34:e8795 and so on). Then authors might mention the advantages of previous reports because the authors analyzed carbon, nitrogen, hydrogen and sulfur.
The readers might have some questions about discriminant analysis. The authors could mention possibility of discriminant analysis of unknown beefs and tracing of geographic origin of beef based on this research.
These analyses seem to be carried out in various sites and the authors mentioned low standard deviations between the sites in line 299. I recommend that the authors might provide more detail information about it as supplement sheet.

Author Response
Reviewer 2:
This manuscript addresses authentication of European beef by using stable isotope analysis. The authors analyze isotope ratio of hydrogen, carbon, nitrogen and sulfur in beef of several regions. The values have been used for earth science such as resource mapping and recently apply for food science. Then, these provide chemical fingerprint which derives from circumstance and scientific evidence. There are various reports about authentication of beef, especially famous and expensive brands. I recommend that this paper be accepted after minor revision.
The isotope ratio of carbon, nitrogen, and hydrogen/oxygen provides difference of feed, origin of protein in feed and isotope ratio of water in growth region (for example, Rapid Commun Mass Spectrom. 2020;34:e8795 and so on). Then authors might mention the advantages of previous reports because the authors analyzed carbon, nitrogen, hydrogen and sulfur.
Thanks to the reviewer for pointing us to this study that we had forgotten. We cited it in the Introduction. In the individual paragraphs of the Results section, we have explained the sources of variability of the various isotopic parameters and the reason why they are investigated (correlation with the diet or with the animal drinking water).
The readers might have some questions about discriminant analysis. The authors could mention possibility of discriminant analysis of unknown beefs and tracing of geographic origin of beef based on this research.
We agree, and we mentioned this possibility in the conclusions.
These analyses seem to be carried out in various sites and the authors mentioned low standard deviations between the sites in line 299. I recommend that the authors might provide more detail information about it as supplement sheet.
We created a supplementary table citing the laboratories involved in the inter-collaborative study and the mean standard deviations obtained.
Reviewer 3 Report
Title: Characterisation of beef coming from different European countries through stable isotope (H, C, N and S) ratio analysis
The manuscript “Characterisation of beef coming from different European countries through stable isotope (H, C, N and S) ratio analysis” suggested that stable isotope measurements highlighted statistical differences between coastal and inland regions, areas sited at different latitudes, regions with different geology and different farming systems, depending on the diet the animals were given (C3-based, C4-based or a mixed one). It is well written article with some interesting findings; however, there are some corrections before its acceptance for publication:
Line 19: delete the word “also”.
Line 19-21: Rephrase the sentence “The analysis of stable isotope ratios….”.
Please add a sentence about materials and methods in the abstract part i.e., how the stable isotope ratio was analyzed.
Line 32-37: Authors should provide reference of the first paragraph of the introduction part. And should refer or discuss some of the studies or review papers from the past.
Line 51-61: References of many sentences are missing, which should be cited.
Line 81: Authors should describe the biochemical mechanism involved behind the influence of photosynthetic pathway that the plants follow for the CO2 fixation on δ(13C)?
Line 127: Correct the percentage “9.6‰”.
Line 149: Authors have discussed about lower levels of δ(34S) in Florence and Sicily, however, what would be possible reason of higher levels of δ(34S) in Chalkidiki (GR), Barcelona (ES), Orkneys and Cornwall (GB).
Line 207: Figure 2: Authors should add adjusted R2 value of the regression analysis?
Line 287: Authors should mention the number and names of the laboratories?
Line 337: Authors should suggest some guidelines for future research, i.e., which aspect should be focused for future research.
English grammar and sentence structure should be revised and corrected throughout the manuscript.
Author Response
Reviewer 3:
The manuscript “Characterisation of beef coming from different European countries through stable isotope (H, C, N and S) ratio analysis” suggested that stable isotope measurements highlighted statistical differences between coastal and inland regions, areas sited at different latitudes, regions with different geology and different farming systems, depending on the diet the animals were given (C3-based, C4-based or a mixed one). It is well written article with some interesting findings; however, there are some corrections before its acceptance for publication:
Line 19: delete the word “also”
We corrected as requested.
Line 19-21: Rephrase the sentence “The analysis of stable isotope ratios….”.
We corrected as requested.
Please add a sentence about materials and methods in the abstract part i.e., how the stable isotope ratio was analyzed.
We added the information as requested.
Line 32-37: Authors should provide reference of the first paragraph of the introduction part. And should refer or discuss some of the studies or review papers from the past.
We added a reference to the first paragraph of the Introduction, but we would prefer to limit the information not concerning the aim of the study to avoid an excessively descriptive introduction.
Line 51-61: References of many sentences are missing, which should be cited.
We added the references, as requested.
Line 81: Authors should describe the biochemical mechanism involved behind the influence of photosynthetic pathway that the plants follow for the CO2 fixation on δ(13C)?
We described the biochemical mechanism involved behind the influence of photosynthetic pathway that the plants follow for the CO2 fixation on δ(13C), as requested.
Line 127: Correct the percentage “9.6‰”.
Thanks for the comment. The value of 9.6‰ is not a percentage, but the maximum nitrogen isotopic value of the dataset, so we should keep the ‰.
Line 149: Authors have discussed about lower levels of δ(34S) in Florence and Sicily, however, what would be possible reason of higher levels of δ(34S) in Chalkidiki (GR), Barcelona (ES), Orkneys and Cornwall (GB).
The sea-spray effect represents a possible reason for these sites to have higher δ(34S) values. We modified the text to highlight this comment.
Line 207: Figure 2: Authors should add adjusted R2 value of the regression analysis?
Thanks for the suggestion. We calculated the adjusted R2 value (0.74), but as it is a lower value than the R2 (0.77), we decided to stick with the latter, as in this way the response variables result to be better explained by the predictor variables.
Line 287: Authors should mention the number and names of the laboratories?
We created a supplementary table citing the laboratories involved in the inter-collaborative study and the mean standard deviations obtained.
Line 337: Authors should suggest some guidelines for future research, i.e., which aspect should be focused for future research.
We added a suggestion for the future development of the research, as requested.
English grammar and sentence structure should be revised and corrected throughout the manuscript.
A native English speaker employed by the company EUROSTREET SOCIETÀ COOPERATIVA (info@eurostreet.it) revised the text.
Round 2
Reviewer 3 Report
Good efforts! The manuscript is sufficiently improved according to the comments and suggestions of the reviewer to accept it in present form for its publication.